# PRISMLAYERS: Open Data for High-Quality Multi-Layer Transparent Image Generative Models

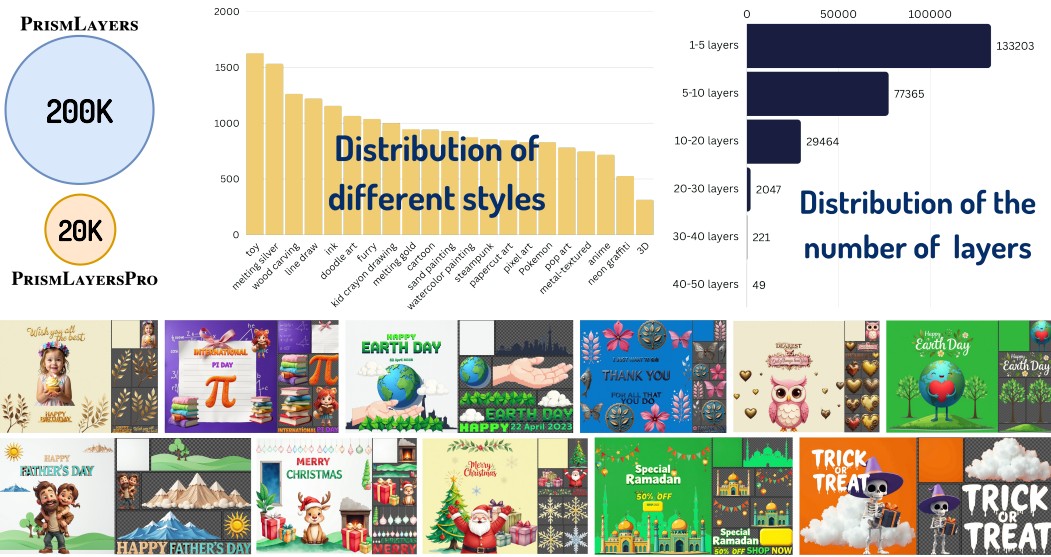

Figure 1: Illustration of key statistics from PRISMLAYERS (number of layers) and PRISMLAYERSPRO (different of styles), along with representative high-quality synthetic multi-layer transparent images from PRISMLAYERSPRO.

## Abstract

Generating high-quality, multi-layer transparent images from text prompts can unlock a new level of creative control, allowing users to edit each layer as effortlessly as editing text outputs from LLMs. However, the development of multi-layer generative models lags behind that of conventional text-to-image models due to the absence of a large, high-quality corpus of multi-layer transparent data. We address this fundamental challenge by: (i) releasing the first open, ultra–high-fidelity PRISMLAYERS (PRISMLAYERSPRO) dataset of 200K (20K) multi-layer transparent images with accurate alpha mattes, (ii) introducing a training-free synthesis pipeline that generates such data on demand using off-the-shelf diffusion models, and (iii) delivering a strong multi-layer generation model, ART+, which matches the aesthetics of modern text-to-image generation models. The key technical contributions include: `LayerFLUX`, which excels at generating high-quality single transparent layers with accurate alpha mattes, and `MultiLayerFLUX`, which composes multiple `LayerFLUX` outputs into complete images, guided by human-annotated semantic layout. To ensure higher quality, we apply a rigorous filtering stage to remove artifacts and semantic mismatches, followed by human selection. Fine-tuning the state-of-the-art ART model on our synthetic PRISMLAYERSPRO

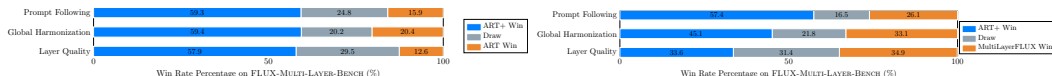

Figure 2: User study results on the effectiveness of PRISMLAYERSPRO. Left: ART+ v.s. ART. Right: ART+ v.s. MultiLayerFLUX. With fine-tuning on PRISMLAYERSPRO, ART+ achieves the best performance.

yields ART+, which outperforms the original ART in 60% of head-to-head user study comparisons and even matches the visual quality of images generated by the FLUX.1-[dev] model. Our work establishes a solid dataset foundation for multi-layer transparent image generation, enabling research and applications that require precise, editable, and visually compelling layered imagery.

🤗 **Dataset:** https://huggingface.co/datasets/artplus/PrismLayersPro

# 1   Introduction

Despite remarkable advances in text-to-image diffusion models, users still face significant challenges in refining outputs to achieve satisfactory results. The difficulty lies in the fact that users cannot precisely articulate their visual requirements before seeing generated images, leading to laborious post-processing workflows. The fundamental issue here is that existing diffusion models are designed to produce single-layer images, lacking the transparent layers and precise alpha mattes required for flexible, layer-wise editing. Modern image editing workflows rely on multi-layered structures for the smooth adjustment of individual elements without causing disruption to the entire composition.

In this paper, we argue for a paradigm shift—from text-to-image generation to text-to-layered-image generation. Such an evolution would empower models to support flexible, layer-wise editing operations that align closely with professional design workflows. The fundamental challenge hindering progress in this area is the lack of high-quality multi-layer image datasets featuring both visually appealing transparency and accurate alpha mattes. Bridging this gap is essential to unlocking the full potential of layered image generation with diffusion models.

Nevertheless, existing literature still relies on the conventional pipeline of fine-tuning generative models on limited, low-quality crawled multi-layer datasets. These datasets have two major drawbacks: (i) aesthetic quality: our empirical analysis shows that the aesthetic scores of crawled multi-layer images are significantly lower than those of RGB images generated by state-of-the-art diffusion models like FLUX.1-[dev]. As a result, we empirically find that fine-tuning on less visually appealing data can degrade the overall aesthetics; (ii) dataset size: the scale of these crawled multi-layer datasets is much smaller than that of conventional RGB image datasets. Consequently, fine-tuning on such datasets becomes less effective as the foundational generative models become increasingly powerful.

This paper leverages off-the-shelf powerful diffusion models to generate high-quality multi-layer transparent images, thereby bypassing the need for fine-tuning on specific datasets. To achieve this goal, this paper makes three key contributions: (i) `LayerFLUX`: We propose a training-free, single-layer transparent image generation system that utilizes a generate-then-matting scheme. Specifically, our approach leverages diffusion models to generate images with solid-colored backgrounds and uses a state-of-the-art image matting model to extract high-quality alpha masks for salient objects. We have named this system `LayerFLUX`, as it builds upon the latest diffusion transformer model, FLUX.1-[dev]. (ii) `MultiLayerFLUX`: We introduce a layout-then-layer scheme that composes multiple high-quality transparent layers generated by `LayerFLUX` according to a given layout, which can be obtained either from a reference image or generated using an LLM. This modular approach enables precise control over spatial composition while preserving the visual quality and alpha matte of each layer, resulting in our `MultiLayerFLUX` system. (iii) `Transparent Image Preference Scoring Model`: We develop a dedicated preference scoring model to assess the visual aesthetics of the generated transparent images. Figure 1 shows the high-quality synthetic multi-layer transparent images generated using `MultiLayerFLUX`.

To demonstrate the effectiveness of above designs, we first compare `LayerFLUX` against previous state-of-the-art transparent image generation methods such as LayerDiffuse [25]. Figure 14 shows the user-study results on a comprehensive benchmark (LAYER-BENCH) that includes prompts describing natural object layers, sticker/text sticker layers, and creative object layers. Second, we leverage `MultiLayerFLUX` to construct a large-scale high-quality multi-layer dataset (PRISMLAYERS) com-

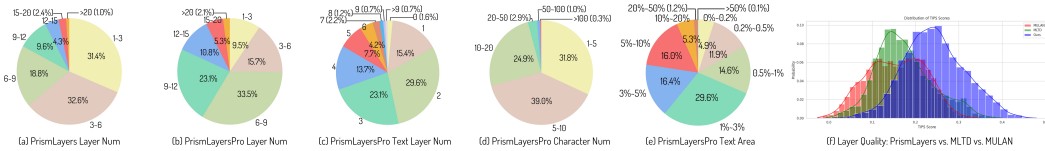

Figure 3: Illustrating the key dataset statistics on PRISMLAYERS and PRISMLAYERSPRO

prising approximately 200K multi-layer transparent images and perform rigid filtering to construct a smaller set of 20K samples with the best quality, forming PRISMLAYERSPRO. We validate the benefits of PRISMLAYERSPRO by fine-tuning the latest multi-layer generation model, ART [19], and present corresponding user-study results in Figure 2, comparing our model's performance with that of the original ART. We find that ART+ is preferred in approximately 57% to 60% of cases across prompt alignment, global harmonization, and layer quality. We empirically find the composed multi-layer images generated with ART+ even match the quality of the holistic single-layer images generated with FLUX.1-[dev] to some extent. These results demonstrate the fundamental role of a high-quality multi-layer transparent dataset in developing the next generation of multi-layer transparent image generation models. We anticipate that our open-source dataset will serve as a solid foundation for future efforts in this direction.

## 2    Related work

Transparent image generation for interactive content is divided into single-layer methods (LayerDiffuse [25], Text2Layer [26], LayeringDiff [11]) and multi-layer methods (LayerDiff [8], ART [19]). Unlike top-down schemes such as MULAN [21], our bottom-up pipeline generates high-fidelity transparent layers before composition, achieving superior aesthetics on PRISMLAYERS. Meanwhile, the graphic-design generation has shifted to business-driven layouts: COLE/OpenCOLE [10, 9] iteratively assembles elements via LLMs and diffusion, and Graphist [6] employs hierarchical layout planning. In this paper, we focus on building an open, high-quality multi-layer transparent image dataset to facilitate future work on closing the gap between multi-layer generation and conventional single-layer text-to-image models. We also discuss the connections and differences between our benchmark and previous multi-layer transparent image generation datasets in Table 1.

## 3    PRISMLAYERS: A High-Quality Multi-Layer Transparent Image Dataset

We introduce PRISMLAYERS, a synthetic dataset consisting of approximately 200,000 multi-layer transparent images. Each sample is accompanied by a global image caption, layer-wise captions, corresponding layer-wise RGB images, and precise alpha mattes. All samples have undergone rigorous aesthetic evaluation and filtering based on our proposed Transparent Image Preference Score (TIPS) model. Furthermore, we curate a high-quality subset of 20,000 images from PRISMLAYERS, termed PRISMLAYERSPRO, representing the top aesthetic tier of the dataset. We will first present detailed statistical characteristics and the curation pipeline of the PRISMLAYERS dataset. Subsequently, we will present our key technical contributions: `LayerFLUX` and `MultiLayerFLUX`.

### 3.1    PRISMLAYERS Statistics

**Statistics on the number of layers.** We analyze the distribution of transparent layer counts in PRISMLAYERS. Each image contains an average of 7 layers (median: 6), with 85% of samples containing between 3 and 14 layers. This indicates that PRISMLAYERS effectively captures a wide range of visual complexity. Figure 3 (a) provides a more detailed illustration of the transparent layer count distribution.

**Statistics on the aesthetics of layers.** A key contribution of this open-source dataset is the provision of aesthetically pleasing transparent layers, addressing the limited visual quality found in existing multi-layer datasets. As shown in Figure 3 (f), quantitative evaluations using our Transparent Image Aesthetic Scoring (TIPS) model illustrate the aesthetic distributions of PRISMLAYERS, MULAN [21], and MLTD [19]. Figure 4 visualizes qualitative comparisons between PRISMLAYERS and PRISMLAYERSPRO.

| Dataset | # Samples | # Layers | Open Source | Source Data | Alpha Quality | Aesthetic |
|---|---|---|---|---|---|---|
| Multi-layer Dataset [25] | $\sim 1$ M | 2 | ✗ | commercial, generated | good | good |
| LAION-L$^2$I [26] | $\sim 57$ M | 2 | ✗ | LAION | normal | normal |
| MLCID [8] | $\sim 2$ M | [2,3,4] | ✗ | LAION | poor | poor |
| MLTD [19] | $\sim 1$ M | $2 \sim 50$ | ✗ | Graphic design website | good | normal |
| MAGICK [5] | $\sim 150$ K | 1 | ✓ | Synthetic | good | good |
| MuLAn [21] | $\sim 44$ K | $2 \sim 6$ | ✓ | COCO, LAION | poor | poor |
| Crello [24] | $\sim 20$ K | $2 \sim 50$ | ✓ | Graphic design website | normal | poor |
| PRISMLAYERS | $\sim 200$ K | $2 \sim 50$ | ✓ | Synthetic | good | good |
| PRISMLAYERSPRO | $\sim 20$ K | $2 \sim 50$ | ✓ | Synthetic | good | excellent |

Table 1: Comparison with previous multi-layer transparent image datasets.

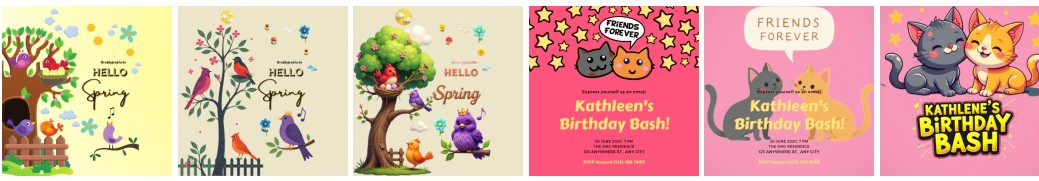

Figure 4: Illustrating the aesthetic quality of the crawled data (columns 1 and 4), synthetic data (columns 2 and 5), and high-quality synthetic data generated with a style prompt (columns 3 and 6).

Our results show that PRISMLAYERS consistently provides higher-quality layers, with the open-source subset PRISMLAYERSPRO achieving the best overall aesthetic quality.

**Statistics of visual text layers.** High-quality visual text rendering is essential for multi-layer transparent image generation, as textual elements play a central role in many business-centric visual designs [16, 17]. PRISMLAYERS contains a large number of accurately rendered text layers, each isolated in a separate transparent channel. Figure 3 (c), (d), and (e) present statistics on the number of text layers per image, the number of characters per instance, and the area ratio of text layers.

**Statistics of different visual styles.** In the middle of Figure 1, we illustrate the distribution of transparent layers across different styles in PRISMLAYERSPRO, which contains 21 distinct styles. The top five most frequent styles are 'toy', 'melting silver', 'line draw', 'ink', and 'doodle art'

**Comparison with existing transparent datasets.** Table 1 presents a comparison with previously existing multi-layer transparent image datasets. We position PRISMLAYERSPRO as the first open, high-quality synthetic dataset that supports a diverse range of layers, high-quality alpha mattes, and excellent aesthetic quality. We believe PRISMLAYERSPRO can serve as a solid foundation for future efforts in building better multi-layer transparent image generation models.

### 3.2 PRISMLAYERS Dataset Curation Process

We illustrate the entire dataset curation pipeline in Figure 5. To ensure clarity, we mark all dataset states with blue-colored markers, including Ⓐ, Ⓑ, Ⓒ, Ⓓ, Ⓔ, and Ⓕ. For the different algorithm operations, we use black-colored markers, including ❶, ❷, ❸, ❹, ❺, and ❻. Further details are explained as follows:

**Multi-layer prompts and semantic layout from crawled data.** Ⓐ → ❶ → Ⓑ We begin by collecting an internal dataset of 800K multi-layer graphic designs sourced from various commercial websites. Each design instance consists of multiple transparent layers, including background elements, decorations, text, and icons. To enrich the semantic understanding of each instance, we employ an off-the-shelf LLM—Llava 1.6 [15]—to generate captions for both individual transparent layers and the fully composed images. This process yields annotations comprising 800K multi-layer prompts and their corresponding semantic layouts, effectively capturing both the visual composition and the intended design semantics. We also extract the original metadata specifying the layer ordering for each graphic. For the filtered PRISMLAYERSPRO set (after ❹), we further enhance semantic richness by using GPT-4o to generate high-quality layer-wise captions.

**Synthetic multi-layer transparent images with MultiLayerFLUX.** Ⓑ → ❷ → Ⓒ With the constructed 800K multi-layer prompts and corresponding semantic layout information, we apply a novel model, MultiLayerFLUX, to transform the layer-wise prompts into multiple transparent

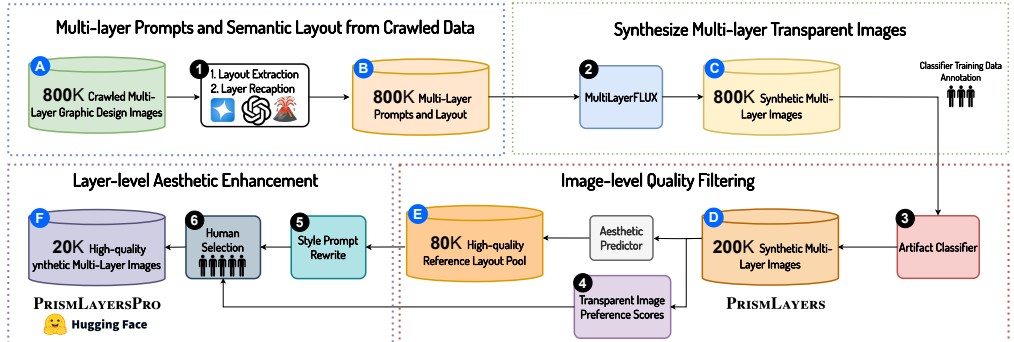

Figure 5: **Dataset Curation Pipeline of PRISMLAYERS and PRISMLAYERSPRO.** We first extract semantic layouts from a database of 800K crawled multi-layer graphic design images. Then, we apply `MultiLayerFLUX` to generate high-quality multi-layer transparent images. An Artifact Classifier is used to evaluate the quality of each composed image, discarding low-quality results to construct PRISMLAYERS. We also apply the Transparent Image Preference Score (TIPS) model to assess the quality of individual transparent layers. By filtering for aesthetic quality and balancing the number of layers, we collect an 80K-image reference layout pool. From this pool, we sample 20K of the highest-quality layouts and regenerate them with style prompts, followed by manual selection—forming our released open-source, high-quality multi-layer dataset, PRISMLAYERSPRO.

layers, each generated separately using a single-layer transparent image generation engine such as `LayerFLUX`, as illustrated in Sec. 3.3. We then composite these transparent layers onto a shared canvas, preserving the correct stacking order and ensuring seamless integration across layers.

A key challenge is that the transparent layers within a multi-layer image often have varying resolutions and aspect ratios. We observe that simply applying `LayerFLUX` to generate each layer within a fixed square canvas tends to produce objects with an unnatural square shape. To remedy this issue, we instead apply `LayerFLUX` to generate transparent layers on canvases that match their original aspect ratios and resolutions.

**Artifact multi-layer transparent image filter. C → ❸ → D** As `MultiLayerFLUX` generates each transparent layer separately and then combines them following the layer order, we observe severe artifacts in some synthetic multi-layer images. These artifacts include duplicate or similar layers positioned in conflicting spatial arrangements or exhibiting substantial and unreasonable overlap, as shown in Figure 6. To address this issue, we construct a reliable artifact classifier to further filter out flawed multi-layer transparent images. We begin by manually annotating severe artifacts in a subset of 8K synthetic multi-layer images with high aesthetic scores. Then, we train an artifact classifier by fine-tuning BLIP-2 [13] to predict confidence scores indicating whether a composed multi-layer transparent image contains such artifacts—e.g., conflicting layer placements or unreasonable overlap. To ensure the quality of the final dataset, we apply the trained classifier to select a subset of 200K synthetic multi-layer transparent images, forming PRISMLAYERS.

**High-quality reference layout pool. D → E** The aforementioned Artifact Classifier performs image-level structural assessment. Next, we perform visual quality filtering using an aesthetic predictor [1]. We rank images with different numbers of layers based on their aesthetic scores, then select a fixed proportion of the highest-scoring images from each group to form an 80K-image high-quality reference layout pool.

**Layer-wise quality filter, styled prompt rewrite, and human selection. E → ❺ → ❻ + ❹ → F** We take three steps to further improve the layer quality: 1) applying a transparent image preference score (TIPS) to evaluate the quality of the transparent layers, 2) rewriting style prompt to enhance the diversity and visual appealing of these transparent layers, and 3) human selection to find the samples with the best quality. We train the TIPS model on a collection of our PRISMLAYERS, single-layer images generated by LayerDiffuse [25], and our reproduction of LayerDiffuse based on FLUX.1-[dev]. We define 20 distinct style keywords, and for each style, we randomly sample 2,000 layouts from the 80K reference layout pool. Each sampled layout's individual layers are pasted onto a gray background and fed to GPT-4o, which rewrites the layer captions to include the target style directives. Next, we employ `MultiLayerFLUX` to regenerate each transparent layer according to its new, style-aware caption. The resulting styled layers are manually reviewed to remove obvious failures

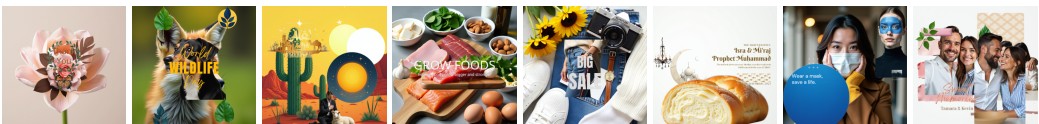

Figure 6: Illustrating the artifact multi-layer transparent images that our classifier can identify and filter out.

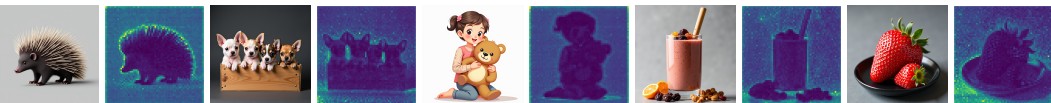

Figure 7: Attention maps between the *suffix text token* and *visual tokens*. We observe a clearly higher attention response in the background area with accurate boundary patterns.

with reference to the scores by our transparent image preference score (TIPS) predictor. Finally, we discard all low-scoring layers or artifact-prone images, producing our final 20K refined high-quality synthetic multi-layer dataset, PRISMLAYERSPRO.

**Discussion.** A natural question is whether the generated multi-layer images exhibit cross-layer coherence. We acknowledge that the synthetic multi-layer transparent images generated by `MultiLayerFLUX` cannot fully guarantee inter-layer consistency. This remains a known limitation of our current scheme, which we mitigate through human selection. Nonetheless, we empirically observe that the recent ART model [19], when trained on our filtered high-quality dataset, produces multi-layer images with noticeably improved coherence—highlighting the value of high-quality supervision in addressing this challenge.

### 3.3 `LayerFLUX` and `MultiLayerFLUX`

In this section, we present the mathematical formulation of the multi-layer transparent image generation task, followed by key insights and implementation details of our `LayerFLUX` and `MultiLayerFLUX` models.

**Formulation.** The transparent image generation task aims to train a generative model that transform the input global text prompt $\mathbf{T}_{\text{global}}$ and the optional regional text prompts $\{\mathbf{T}_{\text{region}}^i\}_{i=1}^N$ into an output consisting of a set of transparent layers $\{\mathbf{I}_{\text{RGBA}}^i\}_{i=1}^N$ that can form a high-quality multi-layer image $\mathbf{I}_{\text{global}}$, and each layer is with accurate alpha channels $\{\mathbf{I}_{\text{alpha}}^i\}_{i=1}^N$. This task degrades to a single-layer transparent image generation task when $N = 1$. Following the latest ART [19], we apply a flow matching model to model the multi-layer transparent image generation task by performing the latent denoising on the concatenation of both the global visual tokens and the regional visual tokens.

**`LayerFLUX`.** As shown in Figure 8, we build the `LayerFLUX` with two key designs, including the suffix prompt scheme and the additional salient object matting to predict the accurate alpha mattes.

Inspired by MAGICK [5], we design a series of tailored suffix prompts to guide diffusion models in generating images with single-colored, uniform backgrounds. These controlled conditions ensure that the foreground elements are clearly delineated, thereby simplifying the isolation process. Our implementation involves simply appending the suffix prompt "*isolated on a gray background*" to the original text prompt. We also compare the results

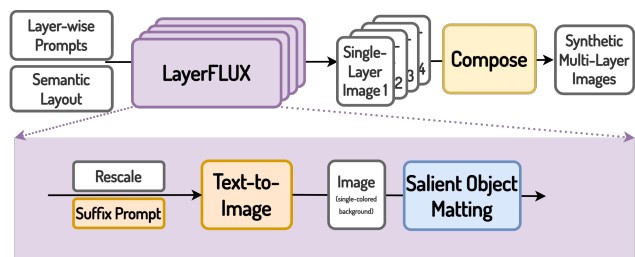

Figure 8: `LayerFLUX` and `MultiLayerFLUX` Framework.

of using alternative suffix prompts by replacing the word "*gray*" with other colors, such as "*green*," "*blue*," "*white*," "*black*," "*half green and half red*," "*half red and half blue*," and others. Figure 7 visualizes the attention maps between the suffix tokens and the visual tokens. We observe that appropriately chosen suffix prompts can guide the diffusion transformer to produce isolated background

regions that are more amenable to segmentation. A detailed analysis of different suffix prompt effects is provided in the supplementary material.

To extract accurate alpha mattes, we explore and evaluate multiple state-of-the-art image matting techniques, including SAM2 [20], BiRefNet [27], and RMBG-2.0 [4], to achieve the separation of the foreground from the background. By leveraging these advanced matting algorithms, we aim for precise alpha matte extraction, ensuring that the edges of the isolated objects are smooth and accurately defined. This step is critical for producing high-quality, transparent images that can be seamlessly integrated into multi-layer compositions. We empirically find that RMBG-2.0 achieves the best matting quality, and we choose it as our default method.

**`MultiLayerFLUX`.** We construct the `MultiLayerFLUX` framework by stacking the outputs from the above-mentioned `LayerFLUX` according to the given layer-wise prompts and semantic layout. Unlike the original FLUX.1-[dev], which directly predicts transparent layers within a square canvas of size $1024 \times 1024$, we preserve the original aspect ratio of each transparent layer and use FLUX.1-[dev] to generate images at varying resolutions, fixing the longer side to $1024$. Each generated transparent layer is then resized to fit the corresponding bounding boxes based on the semantic layout information, and the layers are composited according to the layer-order annotations, resulting in the final synthetic multi-layer transparent images.

## 3.4 Transparent Image Quality Assessment

Existing image quality assessment models [12, 22, 23] are primarily trained to predict human preferences for conventional RGB images, and thus are not well suited for evaluating transparent images with alpha mattes. To address this gap, we propose a dedicated quality scoring model tailored for transparent layer images. The core idea is to distill ensembled preference signals—aggregated from multiple RGB-oriented models—into a model specialized for transparent image quality, thereby mitigating model-specific biases. Furthermore, given that our `LayerFLUX` framework reliably produces high-quality alpha mattes, we exclude explicit transparency-related factors when constructing the preference dataset.

**Transparent image preference dataset.** We first collect a transparent image preference (TIP) dataset of more than 100K win-lose pairs by gathering three types of data resources, including those generated with `LayerFLUX` and LayerDiffuse. We use multiple image quality scoring models to rate the quality of each transparent layer, including Aesthetic Predictor V2.5 [1], Image Reward [23], LAION Aesthetic Predictor [3], HPSV2 [22], and VQA Score [14]. Then, we compare each pair of transparent layers based on the weighted sum of the scores predicted by the aforementioned quality scoring models. Here, we assume that the alpha mask quality of most transparent layers generated with our `LayerFLUX` and LayerDiffuse methods is satisfactory.

**Transparent image preference score.** We train the transparent image preference scoring model by fine-tuning CLIP on the TIP dataset. For each pair of transparent images with preference labels, we choose loss function $\mathcal{L}_{\text{pref}} = (\log 1 - \log \mathbf{p}_w)$, where $\mathbf{p}_w$ is the probability of the win image being the preferred one, and we compute the $\mathbf{p}_w$ as:

$$\mathbf{p}_w = \frac{\exp\left(\tau \cdot f_{\text{CLIP-V}}(\mathbf{I}^w) \cdot f_{\text{CLIP-T}}(\mathbf{T})\right)}{\exp\left(\tau \cdot f_{\text{CLIP-V}}(\mathbf{I}^w) \cdot f_{\text{CLIP-T}}(\mathbf{T})\right) + \exp\left(\tau \cdot f_{\text{CLIP-V}}(\mathbf{I}^l) \cdot f_{\text{CLIP-T}}(\mathbf{T})\right)}, \tag{1}$$

where $f_{\text{CLIP-V}}(\cdot)$ and $f_{\text{CLIP-T}}(\cdot)$ represent the CLIP visual encoder and text encoder separately. $\mathbf{I}^w$ and $\mathbf{I}^l$ represent the prefered and disprefered transparent image.

During the evaluation, we compute the transparent image preference score as follows:

$$\mathbf{p} = f_{\text{CLIP-V}}(\mathbf{I}) \cdot f_{\text{CLIP-T}}(\mathbf{T}), \tag{2}$$

where we directly use the dot product between the normalized CLIP visual embedding and the CLIP text embedding as the transparent image preference score, abbreviated as TIPS for convenience.

## 4 Experiment

### 4.1 Setting

**Implementation details.** We conduct all the experiments with the latest FLUX.1[dev] [2] model. For the fine-tuning of ART [19] on our MultiLayerFLUX datasets, we use 20K training iterations, a

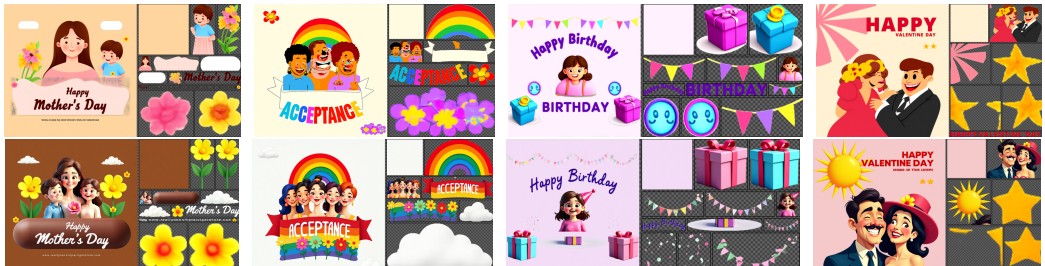

Figure 9: Qualitative comparison results between ART (top row) and ART+ (bottom row).

| Method | DESIGN-MULTI-LAYER-BENCH | | FLUX-Multi-Layer-Bench | |
|---|---|---|---|---|
| | $FID_{merged}$ | TIPS | $FID_{merged}$ | TIPS |
| ART [19] | 18.34 | 16.84 | 30.04 | 16.64 |
| MultiLayerFLUX | 21.29 | 19.90 | 29.64 | 20.65 |
| ART+ | 26.53 | 18.91 | 26.07 | 19.42 |

Table 2: Comparison with state-of-the-art ART.

global batch size of 4, an image resolution of 512×512, and a learning rate of 1.0 with the Prodigy optimizer, followed by fine-tuning at a larger resolution of 1024×1024 with 10K training iterations.

Instead of assessing the model's performance solely on crawled multi-layer graphic designs [19]—most of which follow a similar flat style—we propose evaluating it on a more diverse and creative set generated by the state-of-the-art diffusion model FLUX.1-[dev]. This benchmark is chosen to quantify the gap between generated multi-layer graphic designs and the holistic single-layer image designs produced by the latest text-to-image generation models.

### 4.2 ART+: Improving ART with PRISMLAYERSPRO

**User Study Evaluation.** To assess the effectiveness of our dataset and fine-tuning strategy, we conduct a user study comparing the fine-tuned ART model (denoted as ART+) with the original ART [19], PrismLayers, and PrismLayersPro. Unlike the original ART, which relies on a private multi-layer dataset, we first train ART from FLUX.1-[dev] using the 200K-sample synthetic PrismLayers, and then fine-tune it on the 20K extremely high-quality subset, PrismLayersPro, following the quality-tuning paradigm [7]. The study involves 40 representative samples from FLUX-MULTILAYER-BENCH, with over 20 participants evaluating three key dimensions: (i) *Layer Quality* (visual aesthetics and alpha fidelity), (ii) *Global Harmonization* (inter-layer coherence), and (iii) *Prompt Following* (alignment with input prompts).

As shown in Figure 2, ART+ outperforms the original ART with average win rates of 57.9% in layer quality and 59.3% in prompt following. It also surpasses MultiLayerFLUX in global harmonization (45.1% win rate), validating the impact of combining high-quality supervision with task-specific tuning.

**Quantitative Results.** Table 2 presents the layer-wise TIPS scores and the $FID_{merged}$ scores, comparing the predicted merged images with ground-truth images obtained either from the design test set (DESIGN-MULTI-LAYER-BENCH) or directly from the FLUX image set generated with FLUX.1-[dev]. Our ART+ significantly outperforms ART on the FLUX-MULTI-LAYER-BENCH, and we also provide additional qualitative comparison results below.

**Qualitative MultiLayer Results.** Figure 10 presents qualitative results comparing our MultiLayer-FLUX with the fine-tuned ART+, while Figure 9 shows qualitative comparisons between ART and the fine-tuned ART+. We observe that ART+ achieves significantly better global harmonization than MultiLayerFLUX and better layer quality than ART, separately. These comparisons reveal that the fine-tuned ART+ achieves an excellent balance between layer quality and global harmonization.

**Comparison to FLUX.** Figure 11 compares the merged multi-layer image generation results with the reference ideal images generated directly with FLUX.1-[dev]. We can see that our ART+ significantly outperforms ART and MultiLayerFLUX, achieving aesthetics very close to those of the original modern text-to-image generation models.

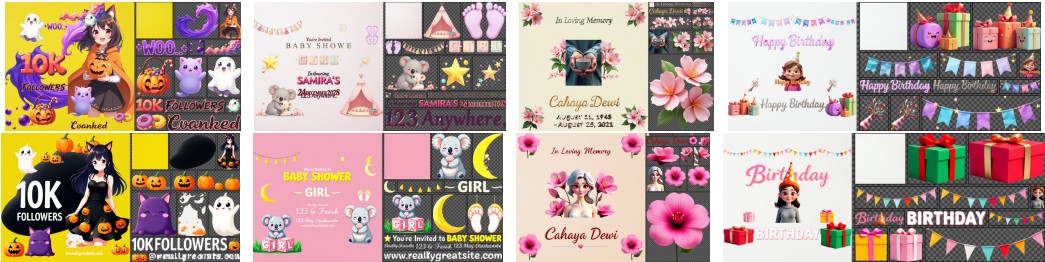

Figure 10: Qualitative comparison results between `MultiLayerFLUX` (top row) and ART+ (bottom row).

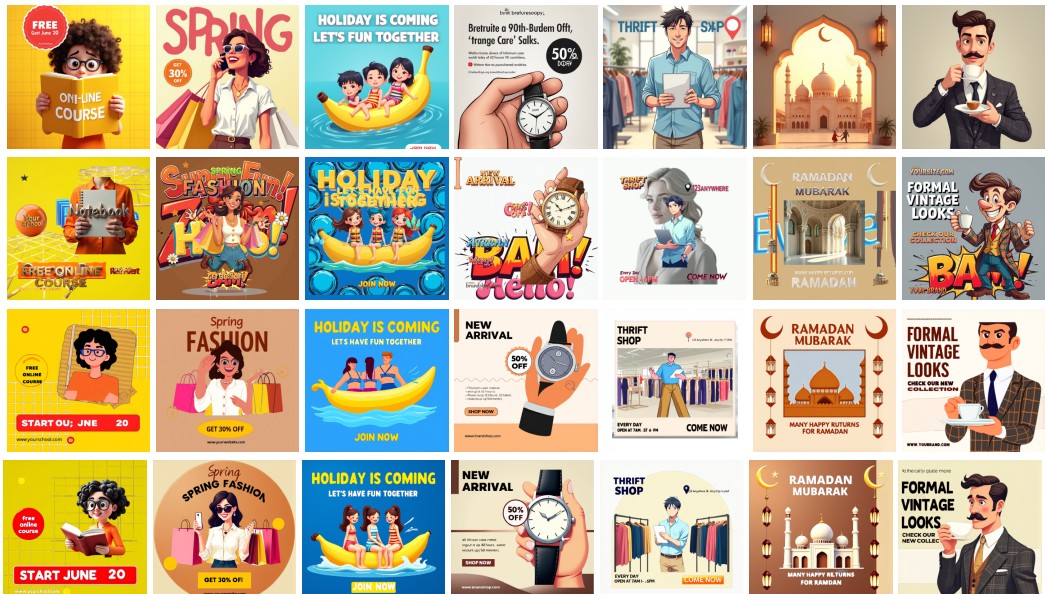

Figure 11: Qualitative comparison results between FLUX.1-[dev] (1st row), `MultiLayerFLUX` (2nd row), ART (3rd row), and ART+ (4th row) across 7 cases (columns). The rightmost columns show composed multi-layer images.

**More Experiments.** We provide more experimental results of `LayerFLUX` and qualitative comparison results in the supplementary materials.

## 5 Conclusion

This paper has tackled the critical gap in multi-layer transparent image generation by assembling and releasing two large-scale datasets—PRISMLAYERS (200K samples) and its ultra-high-fidelity subset PRISMLAYERSPRO (20K samples)—each annotated with precise alpha mattes. To produce this data on demand, we devised a training-free synthesis pipeline that harnesses off-the-shelf diffusion models, and we built two complementary methods: `LayerFLUX` and `MultiLayerFLUX`. After rigorous artifact filtering and human validation, we fine-tuned the ART model on PRISMLAYERSPRO to obtain ART+, which outperforms the original ART in 60% of head-to-head user studies and matches the visual quality of top text-to-image models. By establishing this open dataset, synthesis pipeline, and strong baseline, we lay a solid foundation for future research and applications in precise, editable, and visually compelling multi-layer transparent image generation.

**Limitations & Future Work.** We raise several important questions for future work. How can we generate high-quality multi-layer prompts and semantic layouts without relying on reference data from designers? We observe that even the latest LLMs, including OpenAI o3, still lag behind human-designed layouts and are therefore not yet suitable for multi-layer transparent image generation. How can we generate photorealistic multi-layer transparent images? While PRISMLAYERSPRO focuses on the domain of graphic design, photorealistic images involve more complex inter-layer relationships due to lighting and occlusion effects. We leave these fundamental challenges for future exploration.

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

**A. Details of Suffix Prompt Templates** Table 3 illustrates the detailed suffix prompt templates we adopted for LayerFLUX.

| Method | detailed prompt |
|---|---|
| SuffixPrompt A | on a solid plain gray background. |
| SuffixPrompt B | with a clear, solid gray background. |
| SuffixPrompt C | on a solid single gray background. |
| SuffixPrompt D | floating with a background that is solid gray. |
| SuffixPrompt E | cut-out on a solid gray background. |
| SuffixPrompt F | standing on a background that is fully solid gray |
| SuffixPrompt G | without any surrounding details |
| SuffixPrompt H | isolated on a solid gray background |

Table 3: Effect of choosing different suffix prompt templates.

**B. Generating Multi-Page and Multi-Layer Transparent Slides.** We plan to extend our approach to generate multi-page, multi-layer transparent slides. Our framework not only produces single-layer transparent images but also assembles them into coherent slide decks with multiple pages. Each slide is constructed from several transparent layers, with each layer corresponding to different design elements. This modular, bottom-up strategy enables precise control over both the spatial layout and stylistic attributes of each slide, ensuring consistency across pages while preserving the flexibility to customize individual layers.

**C. Side Effect of Suffix Prompt.** We admit that adding the suffix prompt is not a free lunch and report the results of adding the suffix prompt on the GenEval benchmark in Table 4. We can see that the prompt-following capability of the original text-to-image generation model slightly drops, while the visual aesthetics are maintained.

| Model | Overall | Single | Two | Counting | Colors | Position | Color |
|---|---|---|---|---|---|---|---|
| FLUX.1-[dev] | 0.657 | 0.978 | 0.816 | 0.716 | 0.801 | 0.228 | 0.405 |
| FLUX.1-[dev] + suffix prompt | 0.591 | 0.906 | 0.609 | 0.628 | 0.723 | 0.313 | 0.370 |

Table 4: Comparison results on GenEval.

**D. Technical Details of LayerDiffuse with FLUX.** Our implementation of Layerdiffuse with FLUX is built on FLUX.1-[Dev] with LoRA. Specifically, we convert the image in the MAGICK dataset to grayscale according to the alpha channel mask. After training, the model is capable of generating grayscale background images without the need for additional conditional inputs. Then, we train a transparency VAE decoder to enable the prediction of alpha channels. The decoder is trained on both the MAGICK dataset and an internal dataset, thereby enhancing its robustness and generalization. For the text sticker, we collect a 5k dataset and use GPT-4o to reception of the image.

**E. Experiment Results of LayerFLUX.** We construct a Layer-Bench to evaluate the quality of the single-layer transparent images generated by our `LayerFLUX`. The Layer-Bench consists of 1,500 prompts divided into three types of prompt sets: (i) one that primarily focuses on natural objects sampled from the MAGICK [5] set, where each prompt describes a photorealistic object; (ii) one centers on stickers and text stickers, where the text stickers contains visual text designed in creative typography and style to make the words stand out as part of the visual design; and (iii) one is about creative and stylistic objects. We construct the test set of stickers and text stickers by recaptioning sticker images crawled from the internet.

We compare our approach to the latest state-of-the-art transparent image generation LayerDiffuse [25] by involving more than $\sim 20$ participants from diverse backgrounds in AI, graphic design, art, and marketing. We present system level comparison in Table 6 and the user study results and visual comparisons in Figure 14 and Figure 13. We can see that our LayerFLUX achieves better results across the three types of prompt sets, especially in the creative, stylistic, or text sticker prompt sets. For example, our LayerFLUX achieves better layer quality and prompt following than LayerDiffuse, with win-rates of 63.1% and 61.2% when evaluated on our Layer-Bench. One possible concern might be that LayerDiffuse is built on SDXL [18] rather than FLUX.1-[dev]. We also fine-tune LayerDiffuse on existing transparent image datasets based on FLUX, but we find that the performance is even worse

| # samples | TIPS (Layer Quality) | Composed Image Quality |
|---|---|---|
| Baseline (ART) | 0.114±0.077 | 4.674±0.373 |
| 10 | 0.110±0.076 | 4.684±0.543 |
| 100 | 0.130±0.086 | 4.938±0.418 |
| 1000 | 0.135±0.080 | 4.936±0.415 |

Table 5: Effect of the high-quality data scale.

| Method | Natural Object Layer Quality | | | Sticker Layer Quality | | | Creative Object Layer Quality | | |
|---|---|---|---|---|---|---|---|---|---|
| | HPSv2 ↑ | AE-V2.5 ↑ | TIPS ↑ | HPSv2 ↑ | AE-V2.5 ↑ | TIPS ↑ | HPSv2 ↑ | AE-V2.5 ↑ | TIPS ↑ |
| LayerDiffuse [25] | 26.28 | 5.451 | 29.37 | 21.51 | 3.640 | 19.11 | 29.13 | 5.057 | 32.53 |
| LayerDiffuse w/ FLUX | 24.33 | 5.374 | 27.65 | 25.79 | 4.376 | 25.16 | 25.25 | 4.974 | 29.09 |
| Ours | 26.58 | 5.617 | 30.19 | 26.14 | 4.735 | 25.69 | 29.55 | 5.551 | 36.25 |

Table 6: Comparison with LayerDiffuse on LAYER-BENCH.

than that of the original LayerDiffuse based on SDXL. We infer that *a key reason is that the quality of data generated by these powerful models (like FLUX.1-[dev]) significantly outperforms that of existing transparent images available on the internet or predicted by existing models.* This widening quality gap makes it risky to fine-tune them directly. In summary, our training-free LayerFLUX can better maintain the original capabilities of the off-the-shelf text-to-image generation model, providing a solid foundation for a wide range of applications.

**F. Effect of salient object matting model choice.** How to extract high-quality alpha channels is critical for constructing high-quality single-layer transparent images. We study the influence of different salient object matting models, such as SAM2, BiRefNet, and RMBG-2.0, and summarize the comparison results on LAYER-BENCH in Table 7. We primarily consider the visual aesthetics of the transparent layers after matting and report the quantitative results. Additionally, we visualize the qualitative comparison results in Figure 12. We empirically find that RMBG-2.0 achieves the best results and adopt it as the default model.

**G. Prompt of the Creative Caption Generation** Compared to the common images in the MAGICK dataset, creative images reflect the model's ability to generate less frequent and more novel visual content. To evaluate this capability of our method, we constructed a test set consisting of 500 creative prompts generated by GPT-4o, ensuring diversity and originality in the evaluation dataset. We mainly focus on single objective description generation

**H. Prompt of Multi-layer Style-align Recaption Instruction** Given a reference layer of a multi-layer image, we leverage the visual recognition capabilities of GPT-4o and style-align reception instruction to transfer the original layer caption to a specific style caption. Specifically, we paste the original layer to the center of a gray background image while keeping the aspect ratio. Then, the style-specific instruction and the gray background layer image are fed to GPT-4o. Also, for the generation of ART, we use a similar instruction prompt to transfer the overall writing and style of the global caption.

**I. How to choose the suffix prompt?**

To understand how the suffix prompt helps the transparent layer generation task, we analyze the attention maps between the background regions and the color text tokens within the suffix prompt in Table 8, where we observe that the *"gray"* token achieves the best attention map response. We further conducted a series of experiments to compute mIoU$_{\text{FG}}$ and mIoU$_{\text{BG}}$ by calculating the mean IoU between the binary attention mask and the mask predicted by an image matting model to demonstrate the effect of choosing different suffix prompts quantitatively. In addition, we compute the mean square error between the attention map and the matting mask using MSE$_{\text{BG}}$ and MSE$_{\text{FGLeak}}$, where the latter metric reflects the degree of information leakage from the background to the foreground regions. We compute these metrics as follows:

$$\text{IoU}_{\text{BG}} = \frac{|(1 - \mathbf{M}) \cap \overline{\mathbf{A}}|}{|(1 - \mathbf{M}) \cup \overline{\mathbf{A}}|}, \qquad \text{MSE}_{\text{BG}} = \frac{1}{N} \sum_{i=1}^{N} ((1 - \mathbf{M}_i) - \mathbf{A}_i)^2, \qquad (3)$$

$$\text{IoU}_{\text{FG}} = \frac{|\mathbf{M} \cap (1 - \overline{\mathbf{A}})|}{|\mathbf{M} \cup (1 - \overline{\mathbf{A}})|}, \qquad \text{MSE}_{\text{FGLeak}} = \frac{1}{N} \sum_{i=1}^{N} (\mathbf{M}_i - \mathbf{M}_i \cdot \mathbf{A}_i)^2, \qquad (4)$$

| Method | Natural Object Layer Quality | | | Sticker Layer Quality | | | Creative Object Layer Quality | | |
|---|---|---|---|---|---|---|---|---|---|
| | HPSv2 ↑ | AE-V2.5 ↑ | TIPS ↑ | HPSv2 ↑ | AE-V2.5 ↑ | TIPS ↑ | HPSv2 ↑ | AE-V2.5 ↑ | TIPS ↑ |
| SAM2 | 26.24 | 5.374 | 30.03 | 26.04 | 4.556 | 24.49 | 30.01 | 5.251 | 36.76 |
| BiRefNet | 26.03 | 5.548 | 29.26 | 26.08 | 4.719 | 25.62 | 29.09 | 5.503 | 35.24 |
| RMBG-2.0 | 26.58 | 5.617 | 30.19 | 26.14 | 4.735 | 25.69 | 29.55 | 5.551 | 36.25 |

Table 7: Effect of choosing different salient object matting models.

where $\mathbf{M}$ denotes the binary foreground mask predicted by a state-of-the-art image matting model, and $\overline{\mathbf{A}}$ denotes the binarized version of the attention mask $\mathbf{A}$ computed between the suffix prompt tokens and the visual tokens extracted from the self-attention blocks within the diffusion transformer. $N$ denotes the number of pixels. In addition, we also use a trajectory magnitude to analyze whether the diffusion model is able to control the background region pixels across all timesteps throughout the entire denoising trajectory. Refer to the Appendix for more details.

Figure 7 visualizes the attention maps between the suffix tokens and the visual tokens. We can see that by choosing a suitable suffix prompt, we can elicit the potential of the diffusion transformer to generate isolated background regions that are easy to segment.

| Suffix Prompt | Attention between Suffix text token and visual token | | | | Trajectory Magnitude | |
|---|---|---|---|---|---|---|
| | mIoU$_{BG}$ ↑ | mIoU$_{FG}$ ↑ | MSE$_{BG}$ ↓ | MSE$_{FGLeak}$ ↑ | $\bar{d}_{FG} - \bar{d}_{BG}$ ↑ | $\bar{d}_{BG}$ ↓ |
| original (w/o background prompt) | - | - | - | - | 0.041 | 6.198 |
| half green and half red background | 0.7863 | 0.5943 | 0.4717 | 0.2488 | -0.202 | 6.427 |
| half red and half blue background | 0.7318 | 0.5403 | 0.4868 | 0.2413 | -0.200 | 6.420 |
| half gray and half black background | 0.7902 | 0.5692 | 0.4478 | 0.2468 | 0.243 | 6.062 |
| half gray and half white background | 0.7787 | 0.5540 | 0.4701 | 0.2275 | 0.093 | 6.266 |
| a solid red background | 0.8282 | 0.6398 | 0.4414 | 0.2503 | -1.412 | 7.814 |
| a solid green background | 0.8554 | 0.6646 | 0.4706 | 0.2401 | -0.376 | 6.624 |
| a solid blue background | 0.8379 | 0.6493 | 0.4714 | 0.2416 | -0.485 | 6.818 |
| a solid black background | 0.7318 | 0.5179 | 0.4255 | 0.2409 | -1.749 | 8.317 |
| a solid white background | 0.8070 | 0.6495 | 0.3992 | 0.2365 | -2.503 | 9.083 |
| a solid transparent background | 0.5801 | 0.3302 | 0.4410 | 0.2262 | -1.413 | 7.872 |
| a solid gray background | 0.8642 | 0.6809 | 0.4181 | 0.2564 | 0.805 | 5.591 |

Table 8: Attention-map analysis of different suffix prompts.

**J. Effect of suffix prompt templates.** As shown in Table 8, the design of the suffix prompt is important for guiding the text-to-image generation models to generate images consisting of objects that can be easily isolated from the background by ensuring an approximately single-colored background. Here, we further compare the matting results of nine different suffix prompt designs in Table 9. We empirically find that choosing "*isolated on a solid gray background*" (SuffixPrompt H) achieves slightly better results. We provide the detailed suffix prompts in the appendix.

**K. Effect of *color* within suffix prompt.** One natural question is which color is better for transparent layer generation. We investigate the influence of using different color words within the suffix prompt and summarize the results in Table 10. Accordingly, we find that using the color "gray" achieves the best results. This differs from the observation in previous work [5], which stated that using the color "green" performs best because "green" is the least common hue.

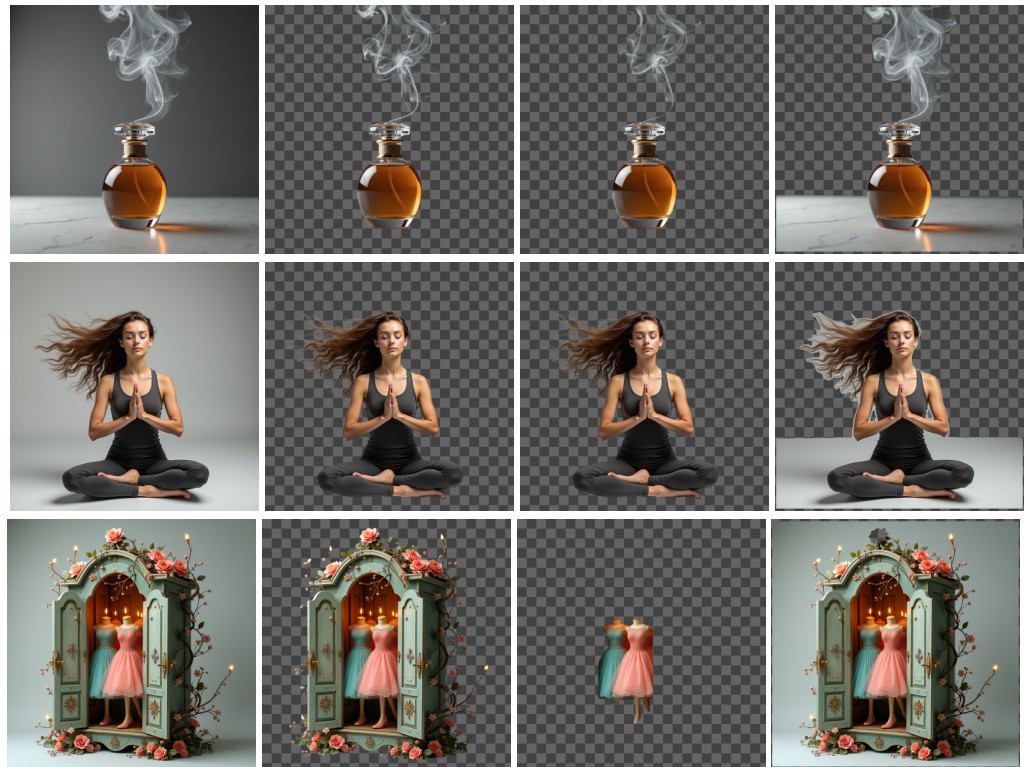

Figure 12: Qualitative comparison of different salient object matting models. From left to right, we show the matted results with RMBG-2.0, BiRefNet, and SAM2.

Table 9: Effect of choosing different suffix prompt templates.

| Method | Natural Object Layer Quality | | | Sticker Layer Quality | | | Creative Object Layer Quality | | |
|---|---|---|---|---|---|---|---|---|---|
| | HPSv2 ↑ | AE-V2.5 ↑ | TIPS ↑ | HPSv2 ↑ | AE-V2.5 ↑ | TIPS ↑ | HPSv2 ↑ | AE-V2.5 ↑ | TIPS ↑ |
| SuffixPrompt A | 26.13 | 5.609 | 29.83 | 26.07 | 4.758 | 25.67 | 29.12 | 5.572 | 36.25 |
| SuffixPrompt B | 26.29 | 5.587 | 29.95 | 25.98 | 4.726 | 25.45 | 29.28 | 5.529 | 36.32 |
| SuffixPrompt C | 26.32 | 5.625 | 30.06 | 26.14 | 4.758 | 25.77 | 29.35 | 5.566 | 36.42 |
| SuffixPrompt D | 25.95 | 5.631 | 29.65 | 26.23 | 4.745 | 25.93 | 29.38 | 5.539 | 36.12 |
| SuffixPrompt E | 26.07 | 5.493 | 29.35 | 26.12 | 4.739 | 25.76 | 28.78 | 5.497 | 34.84 |
| SuffixPrompt F | 26.01 | 5.607 | 29.43 | 26.10 | 4.755 | 25.75 | 29.28 | 5.518 | 35.70 |
| SuffixPrompt G | 26.45 | 5.468 | 30.07 | 25.72 | 4.654 | 25.30 | 29.87 | 5.397 | 36.14 |
| SuffixPrompt H | 26.58 | 5.617 | 30.19 | 26.14 | 4.735 | 25.69 | 29.55 | 5.551 | 36.25 |

Table 10: Effect of choosing different color within suffix prompt.

| Method | Natural Object Layer Quality | | | Sticker Layer Quality | | | Creative Object Layer Quality | | |
|---|---|---|---|---|---|---|---|---|---|
| | HPSv2 ↑ | AE-V2.5 ↑ | TIPS ↑ | HPSv2 ↑ | AE-V2.5 ↑ | TIPS ↑ | HPSv2 ↑ | AE-V2.5 ↑ | TIPS ↑ |
| Gray | 26.58 | 5.617 | 30.19 | 26.14 | 4.735 | 25.69 | 29.55 | 5.551 | 36.25 |
| Green | 25.59 | 5.304 | 28.72 | 25.62 | 4.605 | 25.02 | 28.78 | 5.342 | 34.52 |
| Blue | 26.29 | 5.434 | 29.53 | 25.83 | 4.690 | 25.63 | 29.29 | 5.456 | 35.55 |
| Red | 25.70 | 5.267 | 28.40 | 25.68 | 4.618 | 25.49 | 28.72 | 5.400 | 34.46 |
| White | 24.71 | 4.975 | 27.34 | 25.28 | 4.399 | 24.26 | 27.97 | 5.362 | 34.73 |
| Black | 26.16 | 5.500 | 29.38 | 25.34 | 4.655 | 24.96 | 28.78 | 5.430 | 34.48 |
| Transparent | 26.26 | 5.274 | 29.36 | 25.47 | 4.569 | 24.94 | 29.64 | 5.453 | 36.50 |
| Half green and half red | 25.91 | 5.344 | 29.03 | 25.93 | 4.699 | 26.08 | 29.72 | 5.399 | 35.79 |
| Half red and half blue | 25.83 | 5.418 | 29.10 | 25.99 | 4.691 | 26.05 | 29.75 | 5.459 | 35.89 |

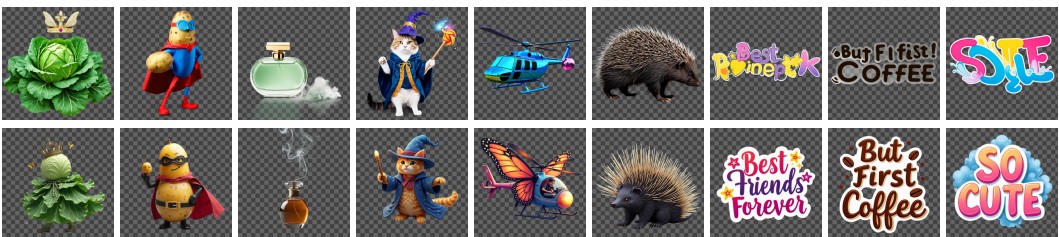

Figure 13: Qualitative comparison of results with SOTA on LAYER-BENCH. The first row shows the results generated with LayerDiffuse, while the second row shows the results generated with our LayerFLUX.

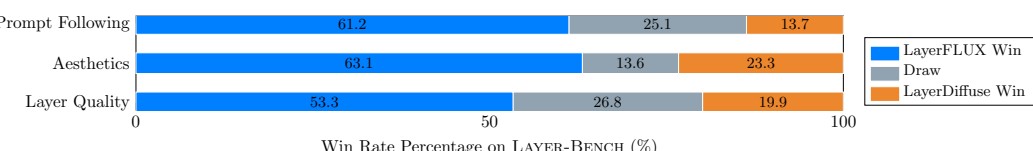

Figure 14: Illustrating the win-rate on single-layer transparent image generation benchmark LAYER-BENCH.

## Text Sticker Recaption Prompt for GPT-4o

You are given the key word of a text sticker and its corresponding image. Your task is to generate an accurate and descriptive caption for the sticker, following these guidelines:
1. The caption begins with "The text sticker describes/contains/" and ends with "isolated on a solid transparent background."
2. Clearly describe the text in the sticker, including the font color, font style, and any visual effects (e.g., shadows, gradients) observed in the image.
3. Keywords usually refer to the text in the sticker, and you may include other relevant descriptive elements. Be explicit about these in your caption.
4. Refer to the examples provided for clarity on how to construct your caption. Aim for creativity while adhering to the required structure.
Here are some examples for reference:
- "The text sticker presents the word 'Focus' in a sharp, modern font, filled with a gradient of charcoal gray to bright red. The letters are outlined in bright white, and stylized targets surround the text, conveying determination and clarity, isolated on a solid transparent background."
- "The text sticker showcases the word 'Celebrate' in a festive, curly font, filled with a vibrant confetti gradient of rainbow colors. Each letter is dotted with tiny sparkles, and balloons and streamers float around, enhancing the joyful spirit of celebration, isolated on a solid transparent background."
Please ensure to generate a caption that fits this style and adheres to the guidelines.
**************************************************
**Response 1**:
{response 1}
**************************************************
Please strictly follow the following format requirements when outputting, and don't have any other unnecessary words.
**Output Format**:
response 1 or response 2.

473

## Creative Object Layer Prompt for GPT-4o

You are tasked with generating imaginative and creative image descriptions based on a given object word. The generated description should follow these specific guidelines:
### **1. Input:**
- You will receive a single object word (e.g., "penguin", "teapot", "robot", etc.).
- Use this object as the central focus of the description.
### **2. Output Requirements:**
- The description should be **creative and unexpected**, modifying the object or adding elements that make it unusual, humorous, or visually striking.
- The description **must not include details about the background**—focus only on the main object and any additional elements that make it more interesting.
- Aim for a **concise but vivid** description, ideally **within 20 to 30 words**.
- Use **strong visual language** to create a mental image.
- Avoid generic descriptions—make it **fun, unique, and imaginative**.
### **3. Examples for Reference:**

| Given Object | Generated Description |
|————|————————|
| Kangaroo | A kangaroo holding a beer, wearing ski goggles and passionately singing silly songs. |
| Car | A car made out of vegetables. |
| Raccoon | A cyberpunk-styled raccoon wearing neon glasses and a futuristic jacket, holding a laser gun in one paw. |
| Teapot | A giant teapot with robotic arms, serving tea while wearing a tiny monocle and top hat. |
| Penguin | A punk-styled penguin with a mohawk, leather jacket, and electric guitar, rocking out on an ice stage. |

### **4. Constraints & Guidelines:**
- Do **not** include the background in the description.
- Feel free to **modify the object's appearance, abilities, or accessories** to make it more interesting.
- If necessary, **add related objects** (e.g., a robot might have futuristic gadgets, a dog might have sunglasses and a skateboard).
- Keep the tone fun, artistic, and engaging.
### **5. Additional Notes:**
Please directly respond to the prompt with the creative description.

474

## Multi-layer Style Recaption Instruction for GPT-4o

You will receive an RGBA image placed on a gray background. Your task is to generate a highly detailed description of the image's content while adhering to a given stylistic (STYLEPROMPT) requirement.

**Key Guidelines:**

1. **Ignore the Gray Background:** - Do not mention or describe the gray background in any way. Focus solely on the foreground content.

2. **Handling Text in the Image:** - If the image contains any textual elements, the description **must** begin with **"Text:"** followed by a precise transcription of all visible text. - Transcribe every word, symbol, punctuation mark, and character **without omission or modification**. - The description of text must be brief and the style description should be limited to 5 words.

3. **Handling Non-Text Elements:** - If the image contains **non-text elements**, generate an **detailed** description, capturing all visible aspects. - Ensure that the provided style, STYLEPROMPT, is seamlessly **integrated into the description**, maintaining coherence and natural flow.

4. **Output Format:** - Provide only the description of the image. Do **not** include any additional explanations, comments, or meta-information about the task itself. - The description **must explicitly state** that the image is in **STYLEPROMPT style**, starting with **"This is a STYLEPROMPT style image."** (VERY IMPORTANT) - Limited to 70 words!!!

The image is shown below:

