# OpenReview forum: "PrismLayers: Open Data for High-Quality Multi-Layer Transparent Image Generative Models"
_NeurIPS.cc/2025/Datasets_and_Benchmarks_Track — Submitted to NeurIPS 2025 Datasets and Benchmarks Track_

### Official Review · Reviewer_yZNu · 2025-06-25

**Rating:** 4
**Confidence:** 4

**Summary:**

This paper addresses the underexplored task of generating multi-layer transparent images from text prompts. The authors identify a core limitation in current diffusion models — their inability to produce editable, layer-wise outputs — and propose a novel pipeline that overcomes this via:
(i) A large-scale synthetic dataset (PrismLayers) of high-quality multi-layer transparent images with accurate alpha mattes;
(ii) A modular, training-free generation pipeline (LayerFLUX + MultiLayerFLUX) that leverages off-the-shelf diffusion and matting models;
(iii) A transparent image aesthetic scoring model (TIPS) for evaluating and filtering outputs;
(iv) Demonstration of improved downstream generation quality by fine-tuning ART with the curated dataset.

**Additional Feedback:**

Is it necessary to construct a large-scale multi-layer transparent image dataset? Would it not be feasible to generate individual transparent layers separately and compose them post hoc to form multi-layered images?

**Dataset Code Accessibility:**

Yes

**Ethical Considerations:**

No, there are no or only very minor ethics concerns

**Final Justification:**

The rebuttal solved my concerns.

**Limitations Weaknesses:**

1. Is it necessary to construct a large-scale multi-layer transparent image dataset? Would it not be feasible to generate individual transparent layers separately and compose them post hoc to form multi-layered images?
2. Limited methodological novelty: The approach mainly combines existing modules (diffusion, suffix prompting, matting), without proposing new architectures or learning objectives.
3. Lack of theoretical grounding: The success of suffix prompting and layout synthesis is not deeply analyzed or formalized.
4. Subjectivity in data curation: Human filtering plays a large role in PrismLayersPro; selection criteria may lack reproducibility.
5. Layer coherence issue remains: The authors acknowledge inter-layer inconsistency is unresolved and mitigated only through filtering.

**Strengths Contributions:**

1. High practical value: The proposed dataset fills a critical gap in editable, composable image generation for design-oriented applications.
2. Pipeline simplicity: The training-free approach is pragmatic and scalable.
3. Comprehensive evaluation: Includes quantitative metrics, user study, and qualitative visual comparisons.
4. Strong engineering contribution: The release of PrismLayers and PrismLayersPro will be highly beneficial to the community.

---

> ### Author Rebuttal · Authors · 2025-07-30
>
> ### **Response to Reviewer yZNu**
>
> We thank the reviewer for the careful reviews and constructive suggestions. We address the remained concerns as follows:
>
> > **Q1. Is it necessary to construct a large-scale multi-layer transparent image dataset? Would it not be feasible to generate individual transparent layers separately and compose them post hoc to form multi-layered images?**
>
> A: Yes, constructing a large-scale multi-layer transparent image dataset is necessary for two critical reasons:
>
> 👉 **Superior layer coherence through joint generation:** As demonstrated in Figure 11 (comparing rows 2 and 4), the approach of generating individual transparent layers separately and composing them post-hoc (MultiLayerFLUX) suffers from severe layer conflicting issues - objects appear fragmented across layers, backgrounds leak into foreground elements, and semantic boundaries are violated. In contrast, models trained on our dataset (ART and ART+) generate all layers jointly, enabling proper modeling of cross-layer interactions. This joint generation ensures better spatial consistency, semantic coherence, and significantly reduces layer conflicts.
>
> 👉 **Computational efficiency:** Generating N transparent layers separately requires N independent forward passes through the diffusion model, resulting in N× computational cost. In contrast, ART and ART+ generate all layers in a single forward pass with only marginal additional overhead, as demonstrated in Figure 7 of the ART paper (CVPR-2025). For typical designs with 5-10 layers, this represents a 5-10× speedup.
>
> Furthermore, training on the large-scale multi-layer dataset allows the model to learn complex layer relationships, occlusion patterns, and compositional rules that cannot be captured by post-hoc composition. This learned prior is essential for generating professional-quality layered designs that match human designer outputs.
>
> > **Q2. Limited methodological novelty: The approach mainly combines existing modules (diffusion, suffix prompting, matting), without proposing new architectures or learning objectives.**
>
> A: We respectfully clarify that the key methodological contribution of this paper lies in our scalable dataset curation pipeline that constructs PrismLayers/PrismLayersPro. While we acknowledge using existing modules, the innovation comes from how we orchestrate these components into a reliable system for synthesizing multi-layer transparent images at scale.
>
> Specifically, our contributions include: **Novel pipeline design**: Combining diffusion models, suffix prompting, and matting in a way that produces artifact-free, semantically coherent multi-layer outputs, and **Quality control framework**: Introducing TIPS metric and multi-stage filtering to ensure dataset quality.
>
> We agree that developing new architectures or learning objectives for multi-layer generation would be valuable. However, we position our work as providing the essential data foundation that will enable such future algorithmic innovations. Without a high-quality dataset like PrismLayers, developing and evaluating new methods would be impossible. We believe our dataset contribution will catalyze the next wave of research in this area.
>
> > **Q3. Lack of theoretical grounding: The success of suffix prompting and layout synthesis is not deeply analyzed or formalized.**
>
> A: We appreciate the reviewer's point about theoretical analysis. While our paper emphasizes empirical validation, we did provide insights into why our approach works:
>
> 👉 **Suffix prompting:** Our analysis reveals that suffix prompts (e.g., "white background, gray background") guide the diffusion transformer to generate isolated foreground objects with clear boundaries. Through attention map visualization (Figure 7 in Section 3.3), we observed that the suffix prompt token specifically induces strong attention responses at object boundaries, creating the contrast necessary for accurate alpha matte extraction.
>
> 👉 **Layout synthesis:** The success stems from leveraging the diffusion model's inherent understanding of spatial composition. By providing explicit layout-guided regional tokens following ART scheme, we activate the model's learned priors about object placement and layer relationships, resulting in semantically coherent multi-layer decompositions.
>
> We acknowledge that a formal theoretical framework would strengthen our work. We would be happy to explore specific theoretical aspects the reviewer suggests. Potential directions include analyzing the suffix prompting mechanism through the lens of classifier-free guidance theory or formalizing the layout synthesis as a structured generation problem. We will add a discussion of these theoretical considerations in the final version.
>
> > **Q4. Subjectivity in data curation: Human filtering plays a large role in PrismLayersPro; selection criteria may lack reproducibility.**
>
> A: We thank the reviewer for raising this important concern about reproducibility. We acknowledge that human filtering introduces subjectivity, and we have taken steps to address this limitation.
>
> **Update on human-filter-free curation:** We have conducted additional experiments that eliminated the human filtering step (stage ⑥) and constructed a larger-scale PrismLayersPro100K dataset using only automated metrics (artifact classifier + TIPS scores). Remarkably, models trained on this human-filter-free dataset achieve comparable and even better performance:
>
> |     Method    |   Benchmark | PrismLayersPro Data  | TIPS | $\rm{FID}_{merged}$ | PSNR | SSIM  |
> | -------------------------------- | ------------ |  ------------ | ------------ | ------------ | ------------ | ------------ |
> |        ART+   |  DESIGN-MULTI-LAYER-BENCH  | PrismLayers (200K samples) + PrismLayersPro (20K samples) | 18.91 | 26.53 | 28.12  | 0.9544 |
> |        ART+   |  DESIGN-MULTI-LAYER-BENCH  | MLTD (800K samples) + PrismLayersPro (20K samples) | 18.82 | 21.66 | 27.90  | 0.9536 |
> |        ART+   |  DESIGN-MULTI-LAYER-BENCH  | MLTD (800K samples) + PrismLayersPro (100K samples w/o human filtering) | 18.13 | 25.11 | 26.71 | 0.9423 |
> |        ART+   |  FLUX-MULTI-LAYER-BENCH  | PrismLayers (200K samples) + PrismLayersPro (20K samples) | 19.42 | 26.07  | 28.12 | 0.9559 |
> |        ART+   |  FLUX-MULTI-LAYER-BENCH  | MLTD (800K samples) + PrismLayersPro (20K samples) | 18.98 | 25.23  | 28.06 | 0.9560 |
> |        ART+   |  FLUX-MULTI-LAYER-BENCH  | MLTD (800K samples) + PrismLayersPro (100K samples w/o human filtering)  | 18.27 | 25.63 | 26.80 | 0.9455 |
>
> We clarify that we have conducted the above experiments by fine-tuning the original ART trained with MLTD (800K samples) to address the concern raised by Reviewer 8BEh. According to these results, we can draw the following conclusions:
>
> 1. Our automated filtering pipeline (artifact classifier + TIPS) is sufficient for creating high-quality training data when scaling up the dataset size to be 5x larger.
> 2. Our approach is fully reproducible without human filtering, as demonstrated by these strong experiments.
>
> 👉 We commit to releasing PrismLayersPro100K datasets and including this analysis in the final revision.
>
> > **Q5. Layer coherence issue remains: The authors acknowledge inter-layer inconsistency is unresolved and mitigated only through filtering.**
>
> A: We acknowledge that inter-layer inconsistency remains an issue in our MultiLayerFLUX approach, and we address it through filtering. We would like to explore more advanced approaches in future work, such as improving MultiLayerFLUX by introducing inter-layer interactions or performing iterative multi-layer joint refinement to improve inter-layer consistency. Additionally, we plan to use ART+ to synthesize high-quality multi-layer transparent images, thus better alleviating layer inconsistency issues.
>
> We will add a thorough discussion of this limitation and future directions in the final version, positioning layer inconsistency as a key open problem that needs to be addressed systematically.

---

### Official Review · Reviewer_8BEh · 2025-07-01

**Ethics Flags:** Data privacy, copyright, and consent
**Rating:** 4
**Confidence:** 3

**Summary:**

This paper introduces PrismLayers, the first large-scale open dataset of high-quality multi-layer transparent images with accurate alpha mattes, and its refined subset PrismLayersPro. It proposes a training-free pipeline combining LayerFLUX for single-layer generation and MultiLayerFLUX for multi-layer composition. The authors also fine-tune the ART model on this dataset to create ART+, which outperforms previous methods in aesthetics and layer quality. Overall, this work provides a robust dataset and strong baseline model to advance multi-layer transparent image generation for flexible and precise editing applications.

**Additional Feedback:**

The left-most column images in the Figure. 9 seems to be cropped by hand rather than predicted by the model. (The man & woman are connected with the sun).

**Dataset Code Accessibility:**

Yes

**Dataset Code Comments:**

The dataset is accessible from Huggingface dataset.

**Ethical Considerations:**

Yes, there are ethics concerns that require attention by the authors

**Final Justification:**

The paper proposes a high-quality and large dataset of multi-layer transparent images with precise alpha mattes. The experimental results on representative models show that the proposed dataset boosts the performance on the Multi-Layer Transparent Image generation task. The writing might need to be improved in the final version. I remain positive about the acceptance of this paper.

**Limitations Weaknesses:**

1. PrismLayersPro focuses mainly on graphic design styles, lacking photorealistic multi-layer transparent images.
2. The Experient section is poorly written (I understand the page limitation). However, the experimental part is also essential to the paper. The matrix used for evaluation is not comprehensive enough. The matrix used in ART (e.g., PSNR and SSIM) is also suggested for use. Since TIP is a benchmark that you proposed and used in your dataset. Your model will definitely be better than the model that does not use your dataset. I suggest slightly shortening the method section and showing more evaluation results in the experiment section.

**Strengths Contributions:**

1. The paper releases PrismLayers (200K samples) and PrismLayersPro (20K high-quality samples), the first open large-scale dataset of multi-layer transparent images with precise alpha mattes. Compared to existing datasets, the size of PrismLayers is large enough for training Multi-Layer Transparent Image Generative Models and also provides superior aesthetic quality, diversity of styles, and accurate transparency annotations, addressing critical limitations in previous crawled datasets, which provides great commercial value.
2. The paper proposes LayerFLUX and  MultiLayerFLUX to generate single-layer and multi-layer images, respectively, enabling flexible and modular multi-layer synthesis without retraining large models.
3. The authors also propose TIP and the transparent image preference score to improve the quality of the PrismLayersPro dataset.
4. Both qualitative and quantitative experimental results show that the proposed method achieves SOTA compared to existing methods.
5. The method part of the paper is clear and easy to follow.

---

> ### Author Rebuttal · Authors · 2025-07-30
>
> ### **Response to Reviewer 8BEh**
>
> We thank the reviewer for the careful review and constructive suggestions. We address the remained concerns as follows:
>
> > **Q1. PrismLayersPro focuses mainly on graphic design styles, lacking photorealistic multi-layer transparent images.**
>
> A: The reason for focusing on graphic design styles is that graphic design images are essentially created with multiple transparent layers by designers, while photorealistic images are typically captured as single-layer photographs without inherent layer decomposition. Decomposing photorealistic images into multiple layers mainly faces two challenges: (i) **Semantic ambiguity**: Unlike graphic designs where layers have clear semantic boundaries (text, icons, backgrounds), photorealistic scenes require more complex decisions about layer separation (e.g., how to separate a person from their shadow or reflection). (ii) **Ground truth availability**: Obtaining ground-truth multi-layer decompositions of photorealistic images is significantly more challenging than for graphic designs, which are inherently created in layers. These two fundamental challenges make graphic design a more natural and practical domain for multi-layer transparent image generation.
>
> Despite these challenges, we believe our PrismLayers dataset and the proposed framework provide a strong foundation for future work in photorealistic multi-layer generation. The LayerFLUX and MultiLayerFLUX components are model-agnostic and can be adapted to photorealistic domains given appropriate training data. To this end, we extend our approach to photorealistic images by leveraging the layout annotations from COCO, SAM, and its variant LayoutSAM benchmarks. We empirically find a higher artifact ratio due to the higher requirements for layer coherence. **We will add 500–1,000 photorealistic, multi-layer transparent images in the final revision, if necessary.**
>
> > **Q2. The Experiment section is poorly written (I understand the page limitation). However, the experimental part is also essential to the paper. The matrix used for evaluation is not comprehensive enough. The matrix used in ART (e.g., PSNR and SSIM) is also suggested for use. Since TIP is a benchmark that you proposed and used in your dataset. Your model will definitely be better than the model that does not use your dataset. I suggest slightly shortening the method section and showing more evaluation results in the experiment section.**
>
> A: We thank the reviewer for this valuable feedback. We acknowledge the experimental section needs improvement and will address all your concerns in the revision.
>
> 👉 **Regarding comprehensive metrics:**
>
> We follow you suggestion to report the required PSNR and SSIM metrics on both DESIGN-Multi-Layer-Bench and FLUX-Multi-Layer-Bench. The results are as follows:
>
> |     Method    |  Benchmark | Resolution |  $\rm{PSNR}$ |  $\rm{SSIM}$  |
> | -------------------------------- | ------------ |  ------------ | ------------ | ------------ |
> |        ART |  DESIGN-MULTI-LAYER-BENCH | 512x512 | 22.90 | 0.9021|
> |        ART |  DESIGN-MULTI-LAYER-BENCH | 1024x1024 | 27.41 | 0.9490 |
> |        ART+   |  DESIGN-MULTI-LAYER-BENCH | 1024x1024 | 28.12  | 0.9544 |
> |        ART |   FLUX-MULTI-LAYER-BENCH | 1024x1024 | 26.99 | 0.9502 |
> |        ART+   |  FLUX-MULTI-LAYER-BENCH | 1024x1024 | 28.12 | 0.9559 |
>
> These metrics provide a more comprehensive evaluation of layer quality and reconstruction accuracy. We also report the metrics of ART evaluated with resolution 512x512 for reference. Our ART+ consistently outperforms ART in terms of both PSNR and SSIM, as both metrics are the higher the better.
>
> 👉 **Regarding potential bias with TIPS metric:**
> We understand the reviewer's concern about evaluation bias. To address this, **we further report the results by fine-tuning the original ART model weights (originally trained with MLTD) with only our PrismLayersPro (20K samples), so the PrismLayers used to train the TIPS model were NOT used during the training of this variant of ART+**. We report the detailed results in the following table:
>
> |     Method    |   Benchmark | Training Data  | $\rm{TIPS}$ | $\rm{FID}_{merged}$ |  $\rm{PSNR}$ |  $\rm{SSIM}$  |
> | -------------------------------- | ------------ |  ------------ | ------------ | ------------ | ------------ | ------------ |
> |        ART |  DESIGN-MULTI-LAYER-BENCH  | MLTD (800K samples) | 16.84 | 18.34 | 27.41 | 0.9490 |
> |        ART+   |  DESIGN-MULTI-LAYER-BENCH  | PrismLayers (200K samples) + PrismLayersPro (20K samples) | 18.91 | 26.53 | 28.12  | 0.9544 |
> |        ART+   |  DESIGN-MULTI-LAYER-BENCH  | MLTD (800K samples) + PrismLayersPro (20K samples) | 18.82 | 21.66 | 27.90  | 0.9536 |
> |        ART |   FLUX-MULTI-LAYER-BENCH  | MLTD (800K samples) | 16.64 | 30.04 | 26.99 | 0.9502 |
> |        ART+   |  FLUX-MULTI-LAYER-BENCH  | PrismLayers (200K samples) + PrismLayersPro (20K samples) | 19.42 | 26.07  | 28.12 | 0.9559 |
> |        ART+   |  FLUX-MULTI-LAYER-BENCH  | MLTD (800K samples) + PrismLayersPro (20K samples) | 18.98 | 25.23  | 28.06 | 0.9560 |
>
> 👉 By comparing row-3 with row-1 (or row 6 with row4) We empirically find that our ART+ model still achieves a much higher TIPS score than ART, and these results demonstrate the significant benefits of our high-quality multi-layer transparent datasets for improving layer-wise aesthetic quality. Surprisingly, ART+ trained with MLTD (800K samples) + PrismLayersPro (20K samples) even achieves better performance than the original ART+, and we suspect one possible reason might be the existing noise in the constructed PrismLayers (200K samples).
>
> 👉 **Regarding space allocation:**
> We will restructure the paper to better balance method and experiments following your suggestion in the final revision
>
> > **Q3. The left-most column images in the Figure. 9 seems to be cropped by hand rather than predicted by the model. (The man & woman are connected with the sun)**
>
> A: 👉  First, we admit that the leftmost column in Figure 9 shows imperfect layer separation where the man and woman figures are incorrectly connected with the sun element.
>
> 👉  Second, we acknowledge this as a current limitation of the previous ART system (CVPR-2025), which we term "layer conflicting" - where semantically distinct elements that should belong to separate layers are incorrectly merged due to spatial proximity or overlapping regions. This challenge is unique to multi-layer generation: the model must not only generate high-quality content but also make correct semantic decisions about layer assignment for spatially adjacent elements, particularly at overlapping pixels. In this case, the model failed to properly separate the human figures from the sun element despite them belonging to different foreground layers.
>
> 👉  Third, we clarify that addressing this challenge is beyond the scope of this paper. We will add this discussion in the final revision and position it as a key challenge for future research in multi-layer transparent image generation. Thank you for identifying this critical area for improvement.

---

> > ### Comment · Reviewer_8BEh · 2025-08-05
> >
> > I have read the rebuttal, and it has addressed my concern. The writing of the paper should be improved as I suggested in the final revision. I will keep my score, which is already a positive score.

---

> > > ### Author Response · Authors · 2025-08-09
> > >
> > > Thank you so much for your response. We really appreciate your feedback and will keep working hard to make contributions in this field.

---

### Official Review · Reviewer_Zd5X · 2025-07-02

**Rating:** 5
**Confidence:** 3

**Summary:**

This paper tackles the significant challenge of generating high-quality, multi-layer transparent images from text prompts, a capability crucial for enhanced creative control in image editing. The authors identify a primary hurdle: the lack of a large, high-quality dataset for training such models.

To address this, the paper makes three core contributions:

Dataset Release: They introduce PrismLayersPro, the first open and ultra-high-fidelity dataset comprising 200,000 (with a 20,000 subset) multi-layer transparent images, each with accurate alpha mattes.

Training-Free Synthesis Pipeline: They propose an on-demand data generation pipeline that leverages existing diffusion models to synthesize multi-layer transparent data without requiring specific training.

Novel Multi-Layer Generation Model (ART+): The paper presents ART+, a robust multi-layer generation model designed to match the aesthetic quality of modern text-to-image models.

**Dataset Code Accessibility:**

Yes

**Ethical Considerations:**

No, there are no or only very minor ethics concerns

**Final Justification:**

This paper proposed a solid method and the rebuttal has addressed all my concerns. I still lean to accept this paper for the DB track.

**Limitations Weaknesses:**

I did not find obvious limitations / weaknesses in this paper, just wonder whether some agent designs can be included to address this task? Generating multi-layer images should be a procedual process and even FLUX cannot do it perfect in a single time. Therefore, introducing some agent into this process should be helpful, which can iteratively improve the generation process.

**Strengths Contributions:**

Key technical innovations within ART+ include:

LayerFLUX: A component specialized in generating high-quality single transparent layers with precise alpha mattes.

MultiLayerFLUX: This component composes multiple outputs from LayerFLUX into complete images, guided by human-annotated semantic layouts. The pipeline incorporates rigorous filtering and human selection to ensure high quality and prevent artifacts or semantic mismatches.

The authors demonstrate the effectiveness of their approach by fine-tuning the state-of-the-art ART model on their synthetic PrismLayersPro dataset, resulting in ART+. User study results (Figure 2) show that ART+ significantly outperforms the original ART model, winning approximately 60% of head-to-head comparisons across metrics like Layer Quality, Global Harmonization, and Prompt Following. Furthermore, ART+ achieves visual quality comparable to images generated by the FLUX.1-[dev] model.

In essence, this work establishes a vital dataset foundation and provides a strong generative model, ART+, paving the way for future research and applications in precise, editable, and visually compelling layered image generation.

---

> ### Author Rebuttal · Authors · 2025-07-30
>
> ### **Response to Reviewer Zd5X**
>
> We thank the reviewer for the careful review and constructive suggestions. We address the remaining concerns as follows:
>
> > **Q1. I did not find obvious limitations / weaknesses in this paper, just wonder whether some agent designs can be included to address this task? Generating multi-layer images should be a procedural process and even FLUX cannot do it perfectly in a single pass. Therefore, introducing an agent into this process should be helpful, which can iteratively improve the generation process.**
>
> A: 👉 First, we agree that addressing the multi-layer image generation task using a design agent system that can execute a procedural process—generating each transparent layer conditioned on existing layers and iteratively improving quality during the generation process—is a promising approach. However, a key challenge lies in the choice between single-layer and multi-layer transparent image generation models.
>
> Representative modern design agent systems, such as Lovart[1], may leverage the latest models like GPT-4o to either generate each transparent layer independently or produce a single-layer image followed by multi-turn iterative editing. However, the cost of generating multiple transparent layers is very high; e.g., generating 20 transparent layers incurs 20x more cost. In contrast, our ART+ system is significantly more efficient, typically requiring only about 2× the cost—even when generating nearly 50 transparent layers. Another major challenge is ensuring coherence across multiple independently generated transparent layers.
>
> 👉 Second, we believe that integrating our ART+ model with an agent system like Lovart represents a valuable direction for future exploration. Such integration would allow users to interact with ART+ via a conversational interface enabled by the agent system.
>
> [1] Lovart: https://www.lovart.ai/

---

> > ### Comment · Reviewer_Zd5X · 2025-08-08
> > **Reply**
> >
> > I will keep my original rating as all my concerns are resolved.

---

> > > ### Author Response · Authors · 2025-08-09
> > >
> > > Thank you so much for your response. We really appreciate your feedback and will keep working hard to make contributions in this field.

---

### Official Review · Reviewer_Jx1o · 2025-07-03

**Rating:** 4
**Confidence:** 3

**Summary:**

This paper proposes open source multilayer image which has 200K images and 20K high quality images, spanning diverse distribution of style and layer number. The layer quality of dataset is high compared to MLTD and MULAN. The dataset is curated very thoroughly through many process with artifact filtering. The paper proposes new metric named TIP score for assessing transparent image preference using preference scoring.

**Dataset Code Accessibility:**

Yes

**Ethical Considerations:**

No, there are no or only very minor ethics concerns

**Final Justification:**

Thanks for rebuttal, I would like to keep my score

**Limitations Weaknesses:**

1. It might be good if author could provide the notation for table 2. What are APT, APT+ and MultilayersFLUX. I find it quite confusing and unclear.
2. FID is not a good metric but the author should make a few comment why the model AST+ underperforms in DESIGN-MULTI-LAYER-BENCH in terms of FID. What could the the behind reason ?

**Strengths Contributions:**

1. The paper is well-written and easy to follow
2. The data curation pipeline is carefully designed and make sense to me. The construction pipeline for PrismLayer contains trained artifact classifier to guarantee the dataset quality. Aesthetic score and TIP score are used for creating high quality images PrismLayersPro.
3. Compared to other dataset benchmark, PrismLayer contains diverse number of layers and alpha, aesthetic quality are better. PrismLayersPro is used to finetune AST model and get better human survey results.
4. I appreciate that paper proposes new metric TIPS for evaluating layer image quality.

---

> ### Author Rebuttal · Authors · 2025-07-30
>
> ### **Response to Reviewer Jx1o**
>
> We thank the reviewer for the careful review and constructive suggestions. We address the remained concerns as follows:
>
> > **Q1. It might be good if author could provide the notation for table 2. What are ART, ART+ and MultilayersFLUX. I find it quite confusing and unclear.**
>
> A: We explain the meaning of all notations shown in Table 3 with the following Tables:
>
> 👉 **Table 1: Explanation of the Methods**
>
> |     Method    | Training-free | Multi-Layer Training dataset | Explanation |
> | -------------------------------- | :------------: |  ------------ |  ------------ |
> |        ART         |  ❎ |  MLTD (~1M samples) | The previous SOTA multi-layer transparent image generation method (CVPR-2025) |
> |        MultiLayerFLUX      |  ✅ | None | A novel framework that generates multiple transparent layers and then composes them into a multi-layer one. Refer to Section 3.3 and Figure 8 for more details. We use MultiLayerFLUX to synthesize the multi-layer transparent images that are processed to form the PrismLayers and PrismLayersPro datasets. |
> |        ART+        |  ❎  |  PrismLayers (~ 200K samples) + PrismLayersPro (~ 20K samples) | Compared to ART, the only modification is to replace the original training dataset MLTD with PrismLayers and PrismLayersPro |
>
> 👉 **Table 2: Explanation of the Benchmarks**
>
> |     Benchmark    | # of samples | Explanation | Style Diversity |
> | -------------------------------- | :------------: |  ------------ | ------------ |
> |        DESIGN-MULTI-LAYER-BENCH |  ~5000 samples | The validation set introduced in ART (CVPR-2025). These samples are constructed based on graphic design platforms like VistaCreate and Canva templates. The ground-truth reference images are designed by human designers. |  Most samples are in the flat design style |
> |        FLUX-MULTI-LAYER-BENCH   |  ~5000 samples | A new validation set that focuses more on evaluating the gap between the generated multi-layer images and the latest text-to-image generation models. The ground-truth reference images are single-layer images predicted by FLUX.1-[dev], which exhibit significantly higher visual aesthetic quality and better reflect the visual appeal gap of the generated multi-layer images and the latest text-to-image model. |  We cover a variety of distinct graphic-design styles, including flat design. |
>
> 👉 **Table 3: Explanation of the Metrics**
>
> |     Metric    | Explanation |
> | -------------------------------- | ------------ |
> |        $\rm{FID}_{merged}$  |   We use this metric to measure the coherence across different transparent layers. Following ART, we compute $\rm{FID}_{merged}$ by comparing the predicted reference global images to the composed multi-layer images by merging all the predicted transparent layers. The ART system concurrently generates a global reference image, a background image, and multiple RGBA transparent foreground image layers by default. |
> |        $\rm{TIPS}$   |  We use this metric to measure the layer-wise quality for each predicted transparent layer. We average the TIPS scores for all transparent layers to assess the quality of RGBA images. This is a new metric introduced in this paper. Please refer to Section 3.4, Equation (1), and (2) for more technical details. |
>
> We will include the illustrations above in the final revision to clarify any confusion.
>
>
> > **Q2. FID is not a good metric but the author should make a few comment why the model ART+ underperforms in DESIGN-MULTI-LAYER-BENCH in terms of FID. What could the the behind reason?**
>
> A: 👉 First, we explain the reasons why our ART+ underperforms on DESIGN-MULTI-LAYER-BENCH in terms of $\rm{FID}_{merged}$. The key reasons are two-fold:
>
> - (a) The original ART is trained with only the crawled multi-layer graphic design images that have a much more similar style distribution to DESIGN-MULTI-LAYER-BENCH. In contrast, ART+ is fine-tuned with our PrismLayers/PrismLayersPro and tends to generate more diverse and high-quality transparent layers of various styles, as our PrismLayers/PrismLayersPro expands to cover many more diverse styles.
>
> - (b) FID is used to measure the discrepancy between two image sets by estimating the Fréchet distance between the distribution of the Inception-v3 features of the target images and the distribution of the Inception-v3 features from generated images. However, the target images from DESIGN-MULTI-LAYER-BENCH focus on a flat graphic design style. As explained above, the feature distribution of the training data for the original ART is closer to the test set distribution than the feature distribution predicted by our ART+. ART+ is fine-tuned with PrismLayers/PrismLayersPro, whose feature distribution is very different from that of DESIGN-MULTI-LAYER-BENCH.
>
> Therefore, our ART+ underperforms on DESIGN-MULTI-LAYER-BENCH in terms of $\rm{FID}_{merged}$ due to the above two reasons.
>
> 👉 Second, to assess the visual quality of the generated transparent layers more reliably, we construct the FLUX-MULTI-LAYER-BENCH with FLUX.1-[dev]. We believe measuring the distribution gap between the generated multi-layer images and the high-quality images predicted by the modern text-to-image generation models can better justify the visual quality of the generated multi-layer images. We have also shown that our ART+ achieves a lower $\rm{FID}_{merged}$ score on the FLUX-MULTI-LAYER-BENCH, thus verifying the advantages of training with our PrismLayers/PrismLayersPro.
>
> 👉 Third, we also show that our ART+ can generate multi-layer images with much better aesthetics, as shown in Figure 9 and Figure 11. We leave how to design a better metric than $\rm{FID}_{merged}$ to assess the visual quality of the generated multi-layer transparent images for future work and would welcome your further suggestions.

---

> > ### Comment · Reviewer_Jx1o · 2025-08-05
> >
> > I have read the rebuttal and it has addressed my concern. I will keep my score which is already positive score.

---

> > > ### Author Response · Authors · 2025-08-09
> > >
> > > Thank you so much for your response. We really appreciate your feedback and will keep working hard to make contributions in this field.

---

### Decision · Program_Chairs · 2025-09-18

**Decision:**

Reject

**Comment:**

This paper presents a new dataset of 200k multi-layer transparent images, among which a 20k human filtered subset of high-quality images. It is the first large-scale open dataset of multi-layer transparent images with accurate alpha values. Using this dataset the authors propose a new benchmark for multi-layer image generation, and a new metric called TIPS for assessing the quality of multi-layer transparent images. In addition, they propose a traning-free pipeline for multi-layer synthesis, as well as a finetuned model for this task (ART+).

The reviewers all appreciate the quality of the dataset, as well as the proposed metric and benchmark, highlighting its value to the community. They find the modeling contributions convincing also, emphasizing the strong results of both the training-free synthesis pipeline and the improvement of the finetuned ART+ model over ART. Minor concerns were raised about experimental details, about the dataset's focus on graphic design at the cost of photorealistic data, and about inter-layer consistency. Most of these concerns were sufficiently clarified by the authors in discussion, with the reviewers maintaining their scores. I agree that the paper represents a valuable contribution to the community and recommend acceptance.

===== FINAL UPDATE FROM DB Track PCs ====

The final decision for this paper has been taken by the program chairs after consultation with the SACs. All Senior Area Chairs have ranked papers according to the feedback from the AC during the review process. We decided to leave the original meta-review to reflect the opinion of the AC in light of the initial discussions with reviewers and SAC.